# Enhanced Photocatalytic Hydrogen Production of ZnIn_2_S_4_ by Using Surface-Engineered Ti_3_C_2_T_x_ MXene as a Cocatalyst

**DOI:** 10.3390/ma16062168

**Published:** 2023-03-08

**Authors:** Mengdie Cai, Xiaoqing Zha, Zhenzhen Zhuo, Jiaqi Bai, Qin Wang, Qin Cheng, Yuxue Wei, Song Sun

**Affiliations:** School of Chemistry and Chemical Engineering, Anhui University, Hefei 230601, China

**Keywords:** Ti_3_C_2_T_x_ MXene, surface functionalization, work function, photocatalytic hydrogen production, cocatalyst

## Abstract

Developing efficient and stable photocatalysts is crucial for photocatalytic hydrogen production. Cocatalyst loading is one of the effective strategies for improving photocatalytic efficiency. Here, Ti_3_C_2_T_x_ (T_x_ = F, OH, O) nanosheets have been adopted as promising cocatalysts for photocatalytic hydrogen production due to their metallic conductivity and unique 2D characterization. In particular, surface functionalized Ti_3_C_2_(OH)_x_ and Ti_3_C_2_O_x_ cocatalysts were synthesized through the alkalization treatment with NaOH and a mild oxidation treatment of Ti_3_C_2_F_x_, respectively. ZnIn_2_S_4_/Ti_3_C_2_T_x_ composites, which were fabricated by the in-situ growth of ZnIn_2_S_4_ nanosheets on the Ti_3_C_2_T_x_ surface, exhibited the promoted photocatalytic performance, compared with the parent ZnIn_2_S_4_. The enhanced photocatalytic performance can be further optimized through the surface functionalization of Ti_3_C_2_F_x_. As a result, the optimized ZnIn_2_S_4_/Ti_3_C_2_O_x_ composite with oxygen functionalized Ti_3_C_2_O_x_ cocatalyst demonstrated excellent photocatalytic hydrogen evolution activity. The characterizations and density functional theory calculation suggested that O-terminated Ti_3_C_2_O_x_ could effectively facilitate the transfer and separation of photogenerated electrons and holes due to the formation of a Schottky junction, with the largest difference in work function between ZnIn_2_S_4_ and Ti_3_C_2_O_x_. This work paves the way for photocatalytic applications of MXene-based photocatalysts by tuning their surface termination groups.

## 1. Introduction

Hydrogen is regarded as an ideal energy with the advantages of a high energy capacity and zero pollutants. Among the various H_2_ production strategies, solar-light-driven photocatalysis for H_2_ production from water splitting is a promising route to alleviating the energy crisis [1,2,3]. Developing highly efficient photocatalysts is the key to realizing the industrialization of photocatalytic H_2_ production. Regarding photocatalysts, ZnIn_2_S_4_ has attracted more attention in recent years because of its low toxicity, visible-light response, and considerable photostability [4,5]. However, the rapid recombination and tardy migration of the photogenerated electrons and holes restricts the photocatalytic H_2_ production efficiency of bare ZnIn_2_S_4_ [6,7]. To address this issue, diverse approaches, including cocatalyst loading, vacancy engineering and heterojunction construction, have been systematically developed to improve the photocatalytic performance of ZnIn_2_S_4_ materials [8,9,10]. Among them, cocatalyst loading has been verified to be a feasible and efficient method to promote the photocatalytic efficiency by accelerating the separation and transfer of photogenerated charge carriers while simultaneously acting as active sites to facilitate the photocatalytic H_2_ production reaction kinetics. The employment of noble metals (such as Pt, Au, Pd and Rh) as cocatalysts, has been proven to be highly efficient in improving the photocatalytic performance, but their high price largely limits their widespread application [11,12]. Therefore, it is urgent to explore an inexpensive and efficient noble metal-free cocatalyst to replace Pt, Au, Pd and Rh to achieve large-scale photocatalytic H_2_ production.

MXene, as an emerging family of 2D transition metal carbides/nitrides, has gained intensive scientific interest in photocatalysis, ascribed to its excellent metal conductivity, large specific surface area with abundant active sites and hydrophilicity [13,14,15]. The 2D planar structure of Ti_3_C_2_T_x_ MXene is beneficial to highly dispersing the host photocatalyst with a strong interfacial contact [16,17,18]. On the other hand, owing to its high conductivity and abundant exposed metal sites, Ti_3_C_2_T_x_ could act as a cocatalyst to facilitate the separation and migration of photogenerated charge carriers and lower the reaction energy barriers for accelerating the reaction kinetics. Therefore, Ti_3_C_2_T_x_ was widely used as a cocatalyst in photocatalytic H_2_ production [18,19,20]. For instance, Zhao et al. [17] reported the construction of hierarchical 2D Bi_2_MoO_6_@Ti_3_C_2_T_x_ by in-situ growing Bi_2_MoO_6_ onto the surface of Ti_3_C_2_T_x_ nanosheets. Ti_3_C_2_T_x_, as the cocatalyst, could not only suppress the agglomeration of Bi_2_MoO_6_ nanosheets and increase the reaction active sites, but also endow the photocatalyst with the Schottky junction. As a result, the Bi_2_MoO_6_@Ti_3_C_2_T_x_ exhibited enhanced photocatalytic activity. Zuo et al. [18] found that the ZnIn_2_S_4_-Ti_3_C_2_T_x_-ZnIn_2_S_4_ sandwich-like hierarchical heterostructures exhibited a superior photocatalytic H_2_ production performance due to the construction of the Schottky junction between ZnIn_2_S_4_ nanosheets and Ti_3_C_2_T_x_. Ran et al. [19] reported that Ti_3_C_2_, as a potential cocatalyst, could efficiently improve the photocatalytic hydrogen production performance by forming the Schottky junction at the Ti_3_C_2_/CdS interface to facilitate the separation of the photogenerated electrons and holes. Meanwhile, they found that the Gibbs free energy for H adsorption (ΔG_H*_) of O-terminated Ti_3_C_2_ is close to zero. With the near-zero ΔG_H*_, the favorable Fermi level position and electrical conductivity, O-terminated Ti_3_C_2_ could serve as an alternative to noble metals in photocatalytic H_2_ production. Liu et al. [20] utilized Ti_3_C_2_ nanosheets acting as the substrate and cocatalyst to synthesize a CdLa_2_S_4_/Ti_3_C_2_ photocatalyst, which could not only promote the dispersion of CdLa_2_S_4_, but also enhance the photogenerated charge carriers separation and transfer, leading to a significant enhancement in photocatalytic H_2_ evolution. In most cases, Ti_3_C_2_ with a large work function could act as electron sink to facilitate the separation and transfer of the photogenerated charge carriers in photocatalytic H_2_ production. In contrast, Peng et al. [21] proposed a dual-carrier-separation mechanism for photocatalytic H_2_ evolution within Cu/TiO_2_@Ti_3_C_2_T_x_, where -OH-terminated Ti_3_C_2_T_x_ with a lower work function than TiO_2_ served as the hole trap to accelerate the holes migration from TiO_2_ to Ti_3_C_2_T_x_. Obviously, the surface termination groups of Ti_3_C_2_T_x_ could arise tunable electronic properties (such as work function) to impact on the photocatalytic performance of the Ti_3_C_2_T_x_-based photocatalysts.

Tailoring the surface termination groups of Ti_3_C_2_T_x_ could alter their work function, electronic and optoelectronic properties [22,23,24]. Recently, the theoretical calculations from Khazaei revealed that the work function of Ti_3_C_2_T_x_ was strongly dependent on the surface termination groups, and the work function of Ti_3_C_2_T_x_ could adjust in a wide range from 1.6 eV to 6.0 eV [24]. Jiang et al. [25] investigated the effect of the surface terminations of Ti_3_C_2_T_x_ on the electrocatalytic H_2_ evolution. They found that O-terminated Ti_3_C_2_T_x_ nanosheets exhibited much higher H_2_ evolution activity than other Ti_3_C_2_T_x_, and the –O termination groups on the basal plane of Ti_3_C_2_ were the H_2_ evolution reaction active sites. Especially, the –O termination groups could promote the adsorption of H and accelerate the H_2_ evolution reaction. However, the insights into the effect of the surface termination groups in Ti_3_C_2_T_x_ MXene-based photocatalysts on the photocatalytic H_2_ production are not established experimentally. Herein, we designed a series of Ti_3_C_2_T_x_ (T_x_ = F, OH, O) with different surface termination groups, and then the 2D ZnIn_2_S_4_ was in-situ grown on the surface of Ti_3_C_2_T_x_ using a facile hydrothermal synthesis method to synthesize ZnIn_2_S_4_/Ti_3_C_2_T_x_ composites. Specifically, the as-synthesized ZnIn_2_S_4_/Ti_3_C_2_O_x_ with the O-terminated Ti_3_C_2_T_x_ exhibited the superior photocatalytic H_2_ production activity. When the content of Ti_3_C_2_O_x_ was 1.0 wt%, the ZnIn_2_S_4_/Ti_3_C_2_O_x_ presented the optimal photocatalytic H_2_ production rate of 363 μmol g^−1^ h^−1^. This work provides us with a paradigm for the rational design of Ti_3_C_2_T_x_ MXene with tailored surface termination groups and the development of efficient MXene-based composites for photocatalytic applications.

## 2. Materials and Methods

### 2.1. Samples Preparation

#### 2.1.1. Synthesis of Ti_3_C_2_F_x_

Typically, 2 g LiF was added into 40 mL HCl aqueous solution (9 M) and stirred for 1 h until the LiF was completely dissolved. A total of 2 g of Ti_3_AlC_2_ powder was then added to the above solution and stirred for 0.5 h. The suspension was stirred at 53 °C for 41 h. Upon cooling, the mixture was centrifuged and washed with deionized water until the pH was close to 7. The product was dried at 60 °C under vacuum for 48 h.

#### 2.1.2. Synthesis of Surface Functionalized Ti_3_C_2_T_x_

In order to obtain the surface functionalized Ti_3_C_2_T_x_, the pristine Ti_3_C_2_F_x_ were treated with a different functionality-modification strategy. To achieve Ti_3_C_2_(OH)_x_ with −OH rich termination groups, according to the previous literature [25], 0.2 g of the pristine Ti_3_C_2_F_x_ was dispersed in 100 mL of 1 M NaOH aqueous solution in order to replace the −F surface termination groups with −OH. After stirring for 2 h at room temperature, the product was centrifuged and washed with deionized water until the pH was close to 7. Then, the product was collected and dried at 60 °C under vacuum for 12 h. To obtain O-terminated Ti_3_C_2_O_x_, the Ti_3_C_2_F_x_ was calcined under 300 °C in Ar gas flow for 2 h.

#### 2.1.3. Synthesis of ZnIn_2_S_4_ and ZnIn_2_S_4_/Ti_3_C_2_T_x_

Typically, 0.176 g InCl_3_·4H_2_O, 0.041 g ZnCl_2_ and 0.120 g thioacetamide (TAA) were added consecutively into 40 mL glycerol aqueous solution (20 vol%) and stirred for 0.5 h. The quantitative Ti_3_C_2_T_x_ (T_x_ = F, OH, O) (1.0 wt%) was added into the above solution. The mixed suspension was heated at 80 °C with stirring for 2 h. After centrifugation, the products were collected and dried at 60 °C for 12 h.

For comparison, the preparation of the pristine ZnIn_2_S_4_ was similar to that of ZnIn_2_S_4_/Ti_3_C_2_T_x_ without the introduction of Ti_3_C_2_T_x_.

### 2.2. Photocatalytic H_2_ Production Experiments

The photocatalytic H_2_ production tests were carried out in a Pyrex glass reaction (Beijing Perfectlight Labsolar-6A, Perfectlight Technology, Beijing, China) with a circulated cooling water system to maintain the temperature at 8 °C. A total of 100 mg of the photocatalyst was dispersed in 100 mL aqueous solution, containing 10 vol% triethanolamine (TEOA) as the sacrificial agent. Before irradiation under a Xe lamp (CEL-HXUV300, Perfectlight Technology, Beijing, China), the suspension was evacuated by the vacuum pump. The produced H_2_ volume was analyzed using an on-line gas chromatograph (GC5190, TCD, A column, Ar carrier).

### 2.3. Characterization

The powder X-ray diffraction (XRD) patterns of the prepared samples were collected using a Rigaku SmartLab (9 kW, Tokyo, Japan) diffractometer with Cu Kα radiation (λ = 0.15418 nm) operating at 40 kV and 4 mA. The morphological analysis of the samples were recorded with scanning electron microscopy (SEM) using a Regulus 8230 scanning electron microscope (Hitachi, Ltd., Tokyo, Japan) at an acceleration voltage of 5 kV. X-ray photoelectron spectroscopy (XPS) was carried out to investigate the surface chemical environment of the samples using an Escalab 250Xi X-ray photoelectron spectrometer (Thermo Fisher Scientific, Waltham, MA, USA). Measuring with an ultraviolet photoelectron spectrometer (Thermo Fisher Scientific, Waltham, MA, USA) was performed with a −5 V bias voltage. The data were calibrated with a C1s spectrum of 284.6 eV. The Fourier transform spectrophotometer (Vertex80 + Hyperion2000, Bruker, Billerica, MA, USA) was employed to acquire IR spectra with the standard KBr disk method. Transmission electron microscopy (TEM), high resolution transmission electron microscopy (HRTEM) images and selected area electron diffraction (SAED) patterns of the samples were collected with a field-emission electron microscope (JEM-2100F, JEOL, Tokyo, Japan). UV-visible diffuse reflectance spectroscopy (UV-vis DRS) was recorded to study the optical absorption ability of photocatalysts with Hitachi U-4100 UV-visible spectrometer using a reference standard of BaSO_4_. The photoluminescence (PL) spectra and time-resolved fluorescence spectra were conducted on an Edinburgh FLS 1000 spectrometer (Edinburgh Instruments Ltd., Livingstone, UK) over an exaction wavelength of 375 nm. Electrochemistry impedance spectroscopy (EIS), Mott–Schottky analyses and transient photocurrent spectra were measured using a CHI660E analyzer (CH Instruments, Inc., Bee Cave, TX, USA) with a standard three-electrode system.

## 3. Results

### 3.1. Schematic Illustration of the Synthesis

The schematic illustration in Figure 1 shows the synthesis process for the ZnIn_2_S_4_/Ti_3_C_2_T_x_ (T_x_ = F, OH, O) samples, which consists of three steps: the preparation of the Ti_3_C_2_ by the selective etching of Ti_3_AlC_2_, surface post-treatment (the alkalization treatment with NaOH and the mild oxidation treatment with Ar calcination) of Ti_3_C_2_F_x_ to replace the –F termination groups with –OH or –O groups and the in-situ hydrothermal synthesis of ZnIn_2_S_4_ on surface of Ti_3_C_2_T_x_.

### 3.2. Samples Characterization

The X-ray diffraction (XRD) patterns of Ti_3_AlC_2_ and the as-prepared Ti_3_C_2_T_x_ (T_x_ = F, OH, O) samples in Appendix A demonstrated a typical Ti_3_AlC_2_ and Ti_3_C_2_T_x_ MXene phase. No crystal structure variation was observed for the Ti_3_C_2_(OH)_x_ and Ti_3_C_2_O_x_, indicating that the surface functionalization treatments just modulated the termination groups without changing the crystalline structure of Ti_3_C_2_F_x_. The XRD pattern of Ti_3_C_2_O_x_ showed no peaks of TiO_2_. Meanwhile, the morphology of the Ti_3_C_2_T_x_ nanosheets was maintained even after the alkalization and oxidation treatments (Appendix A).

To confirm the surface termination groups of the as-prepared Ti_3_C_2_T_x_ (T_x_ = F, OH, O) samples, we performed X-ray photoelectron spectroscopy (XPS), as shown in Figure 1a–c. Figure 1a showed the high-resolution XPS spectrum of F 1s, the binding energy at 685.8 eV was assigned to the Ti-F bond [26]. After the alkalization treatment and mild oxidation treatment of Ti_3_C_2_F_x_, the Ti-F peak intensity in Ti_3_C_2_(OH)_x_ and Ti_3_C_2_O_x_ both significantly decreased, indicating that the surface functionalization treatments did not change its crystal structure, while the termination groups had modulated noticeably. The elemental composition result determined by XPS (Appendix A) also confirmed the decrease of the –F termination groups. As seen from the Ti 2p XPS spectra in Figure 1b, more detailed structural variation could be obtained, four doublets were fitted for Ti 2p_3/2_ and Ti 2p_1/2_, which indicated that the Ti species in Ti_3_C_2_T_x_ exhibited four kinds of chemical environment. The Ti 2p_3/2_ binding energies at approximately 455.1, 455.8, 456.9 and 459.1 eV could be assigned to C-Ti-C, C-Ti-OH, C-Ti-O and O-Ti-O bonds, respectively [23,27,28]. Obviously, compared to Ti_3_C_2_F_x_, the intensity of the C-Ti-O peak for Ti_3_C_2_O_x_ increased and the intensity of the C-Ti-OH peak for Ti_3_C_2_(OH)_x_ increased, which indicated that the –F terminations in the Ti_3_C_2_F_x_ were replaced by –O and –OH after the oxidation treatment and alkalization treatment, respectively. The intensity of the O-Ti-O peak increased in Ti_3_C_2_O_x_ and Ti_3_C_2_(OH)_x_, which was attributed to the surface oxidation with the transform C-Ti-C band to O-Ti-O. Furthermore, the O 1s XPS spectra (Figure 1c) exhibited Ti-O, Ti-OH and C-OH peaks at the binding energies of 530.1, 531.8 and 533.5 eV, respectively [29,30]. In particular, the peak at 531.8 eV demonstrated the highest proportion of –OH groups on the surface of Ti_3_C_2_(OH)_x_, while Ti_3_C_2_O_x_ showed the highest concentration of Ti-O due to O-terminated surfaces. The XPS results showed the coexistence of Ti-F, Ti-OH and Ti-O bonds in all Ti_3_C_2_T_x_ samples. It should be noted that the Ti_3_C_2_O_x_, Ti_3_C_2_(OH)_x_ and Ti_3_C_2_O_x_ represented a higher density of termination groups –F, –OH and –O on the surface, respectively. After the alkalization treatment, the Ti-F peak intensity significantly decreased while the Ti-OH peak intensity increased in Ti_3_C_2_(OH)_x_, implying the successful replacement of –F with –OH. Similarly, the –F groups in Ti_3_C_2_F_x_ were successfully replaced by –O with the mild oxidation treatment to form the Ti_3_C_2_O_x_.

The surface termination groups of the Ti_3_C_2_T_x_ samples were further analyzed using Fourier transform infrared spectroscopy (FTIR), as displayed in Figure 2d. The FTIR spectrum of Ti_3_C_2_T_x_ samples showed two peaks at approximately 3430 and 1625 cm^−1^, which assigned to the −OH band on the surface of Ti_3_C_2_T_x_. In addition, a peak at 657 cm^−1^ can be observed, which is attributed to the Ti-O band [31]. It is notable that Ti_3_C_2_(OH)_x_ showed the strongest −OH vibration intensity and that the Ti_3_C_2_O_x_ exhibited a significantly increased Ti-O vibration, which was consistent with the XPS results. These results indicated that the surface functionalized Ti_3_C_2_(OH)_x_ and Ti_3_C_2_O_x_ were successfully synthesized with the alkalization treatment and mild oxidation treatment, respectively.

The ZnIn_2_S_4_/Ti_3_C_2_T_x_ (T_x_ = F, OH, O) composites were obtained through the in-situ growth of ZnIn_2_S_4_ onto the surface of Ti_3_C_2_T_x_. To acquire the crystallinity phase of the ZnIn_2_S_4_ and ZnIn_2_S_4_/Ti_3_C_2_T_x_ composites, the XRD analysis was introduced (Appendix A). It was found that all ZnIn_2_S_4_/Ti_3_C_2_T_x_ samples presented similar diffraction peaks with ZnIn_2_S_4_. The missing Ti_3_C_2_T_x_ diffraction peaks could be ascribed to the low content and high dispersion of Ti_3_C_2_T_x_ in the composites. The morphology of the ZnIn_2_S_4_ and ZnIn_2_S_4_/Ti_3_C_2_O_x_ samples were investigated using scanning electron microscopy (SEM). The ZnIn_2_S_4_ presented a morphology of nanoflowers stacked with nanosheets (Appendix A). From the SEM image of ZnIn_2_S_4_/Ti_3_C_2_O_x_ sample in Figure 3a, it can be seen that the ZnIn_2_S_4_ particles are uniformly dispersed and anchored onto the Ti_3_C_2_O_x_ surface. The more detailed microstructure of the ZnIn_2_S_4_/Ti_3_C_2_O_x_ composite were further demonstrated using the transmission electron microscopy (TEM) technique. TEM observation confirmed such hierarchical ZnIn_2_S_4_/Ti_3_C_2_O_x_ structure (Figure 2b). Furthermore, as shown in Figure 2c, the lattice distances of the ZnIn_2_S_4_/Ti_3_C_2_O_x_ photocatalyst were measured, and the lattice fringes spacing of 0.32 and 0.41 nm were corresponded to the (102) and (006) planes of ZnIn_2_S_4_, while the lattice fringes spacing of 0.26 nm was assigned to the (0110) crystal plane of Ti_3_C_2_O_x_. Moreover, there was an obvious interface contact observed between the ZnIn_2_S_4_ and the Ti_3_C_2_O_x_, which could contribute to the faster transfer of the photogenerated charge. In addition, the corresponding EDX elemental mapping (Figure 2d) displayed that the Zn, In, S, Ti and C elements were uniformly distributed in the ZnIn_2_S_4_/Ti_3_C_2_O_x_ sample. The above results powerfully confirmed that the ZnIn_2_S_4_/Ti_3_C_2_O_x_ photocatalyst was successful constructed.

The optical properties of pristine ZnIn_2_S_4_ and ZnIn_2_S_4_/Ti_3_C_2_F_x_ (T = F, OH, O) composites were analyzed using the UV-vis diffuse reflectance spectra (UV-vis DRS). As shown in Figure 3a, the pristine ZnIn_2_S_4_ showed an absorption edge at 560 nm, while the absorption edge of the ZnIn_2_S_4_/Ti_3_C_2_T_x_ composites exhibited a slightly red shift with the introduction of Ti_3_C_2_T_x_. Moreover, compared to that of ZnIn_2_S_4_, the absorption intensities of the ZnIn_2_S_4_/Ti_3_C_2_T_x_ composites increased in the whole visible light region, suggesting that the Ti_3_C_2_T_x_ loading increased the visible light utilization efficiency of ZnIn_2_S_4_. In addition, the UV-vis DRS spectra of ZnIn_2_S_4_ and ZnIn_2_S_4_/Ti_3_C_2_T_x_ composites were converted into Tauc’s band gap plots (Figure 3b), the band gaps of ZnIn_2_S_4_, ZnIn_2_S_4_/Ti_3_C_2_F_x_, ZnIn_2_S_4_/Ti_3_C_2_(OH)_x_ and ZnIn_2_S_4_/Ti_3_C_2_O_x_ were measured to be 2.64 eV, 2.60 eV, 2.59 eV and 2.63 eV, respectively.

### 3.3. Photocatalytic H_2_ Evolution Activity

The photocatalytic H_2_ evolution activity of the as-obtained pure ZnIn_2_S_4_ and ZnIn_2_S_4_/Ti_3_C_2_T_x_ composites were evaluated under visible light irradiation using triethanolamine (TEOA) as a sacrificial reagent. It was well known that Ti_3_C_2_T_x_ were not semiconductors and they could not generate electrons and holes upon light irradiation [32]. Therefore, Ti_3_C_2_T_x_ had no photocatalytic H_2_ evolution activity. In Figure 4a, the pure ZnIn_2_S_4_ exhibited the poor H_2_ evolution rate of 253 μmol h^−1^ g^−1^. Inspiringly, after loading the Ti_3_C_2_T_x_ cocatalysts, the ZnIn_2_S_4_/Ti_3_C_2_T_x_ composites all exhibited the improved photocatalytic H_2_ evolution activity, and the order of photocatalytic activity was ZnIn_2_S_4_/Ti_3_C_2_O_x_ > ZnIn_2_S_4_/Ti_3_C_2_F_x_ > ZnIn_2_S_4_/Ti_3_C_2_(OH)_x_ > ZnIn_2_S_4_. Furthermore, the photocatalytic H_2_ evolution rate of the ZnIn_2_S_4_/Ti_3_C_2_O_x_ composites strongly depended on the amount of Ti_3_C_2_O_x_. The ZnIn_2_S_4_/Ti_3_C_2_O_x_ composite with 1.0 wt% Ti_3_C_2_O_x_ achieved the optimal H_2_ evolution rate of 363 μmol h^−1^ g^−1^ (Figure 4b). By further increasing the Ti_3_C_2_O_x_ content, the H_2_ evolution rate of the ZnIn_2_S_4_/Ti_3_C_2_O_x_ composite decreased, which could be due to the excessive amount of Ti_3_C_2_O_x_ covering the active sites and hindering the light absorption of ZnIn_2_S_4_ [33]. The photocatalytic stability test of ZnIn_2_S_4_/Ti_3_C_2_O_x_ for photocatalytic H_2_ evolution was carried out for four consecutive cycles (Figure 4c). It can be seen that ZnIn_2_S_4_/Ti_3_C_2_O_x_ maintained the photocatalytic H_2_ evolution activity during the four consecutive cycles, indicating the excellent photostability of ZnIn_2_S_4_/Ti_3_C_2_O_x_.

### 3.4. The Mechanism of Enhanced Photocatalytic Activity

To shed light on the fundamental reasons for the enhanced photocatalytic performance of ZnIn_2_S_4_/Ti_3_C_2_O_x_, fluorescence property and photoelectrochemical measurements were employed. It is well known that the transfer efficiency of photogenerated electrons and holes was an important influencing factor for the photocatalytic performance. The photoluminescence (PL) spectrum was employed to illustrate the transfer efficiency of the photogenerated electrons and holes. Figure 5a showed the PL spectra of the ZnIn_2_S_4_ and ZnIn_2_S_4_/Ti_3_C_2_T_x_ composites measured at 375 nm. The order of the PL signal intensity at 565 nm was ZnIn_2_S_4_ > ZnIn_2_S_4_/Ti_3_C_2_(OH)_x_ > ZnIn_2_S_4_/Ti_3_C_2_F_x_ > ZnIn_2_S_4_/Ti_3_C_2_O_x_. The loading of Ti_3_C_2_T_x_ lead to the decreased PL intensity of ZnIn_2_S_4_, and the ZnIn_2_S_4_/Ti_3_C_2_O_x_ composite showed the lowest PL intensity, which indicated that the addition of the Ti_3_C_2_O_x_ cocatalyst could effectively facilitate the transfer of the photogenerated electrons and hole on the ZnIn_2_S_4_ photocatalyst. The time-resolved photoluminescence (TRPL) spectra (Figure 5b) further certified this result. The calculated average fluorescence lifetime (Ave. τ) of ZnIn_2_S_4_/Ti_3_C_2_O_x_ (0.594 ns) was significantly longer than that of ZnIn_2_S_4_ (0.167 ns), which demonstrated that the Ti_3_C_2_O_x_ cocatalyst loading greatly reduced the recombination rate of the photogenerated electrons and holes on ZnIn_2_S_4_. In addition, electrochemical impedance spectroscopy (EIS) and transient photocurrent response analyses were carried out to further investigate the separation and transfer ability of the photogenerated charge carriers. The EIS Nyquist plots were shown in Figure 5c and the arc radius on the EIS Nyquist plot of ZnIn_2_S_4_/Ti_3_C_2_O_x_ was the smallest among these four samples, which indicated its lowest resistance for the charge carriers on the ZnIn_2_S_4_/Ti_3_C_2_O_x_ composite. This also confirmed that the Ti_3_C_2_O_x_ cocatalyst enhanced the separation and transfer efficiency of the photogenerated electrons and holes of ZnIn_2_S. The transient photocurrent densities of the as-prepared samples were displayed in Figure 5d. Compared with that of the blank ZnIn_2_S_4_, the photocurrent densities of the ZnIn_2_S_4_/Ti_3_C_2_T_x_ samples exhibited remarkable increases; in particular, ZnIn_2_S_4_/Ti_3_C_2_O_x_ exhibited the highest photocurrent density, further confirming the excellent photogenerated carriers transfer and separation ability of ZnIn_2_S_4_/Ti_3_C_2_O_x_. All of these results proved that the ZnIn_2_S_4_/Ti_3_C_2_O_x_ exhibited the fastest transfer and separation ability of photogenerated electrons and holes, further resulting in the excellent photocatalytic H_2_ production performance.

In terms of the band theory, electron transfer behavior is closely related to the work functions of ZnIn_2_S_4_ and Ti_3_C_2_T_x_ (T_x_ = F, OH, O). In order to determine the work functions (Φ) of the ZnIn_2_S_4_ andTi_3_C_2_T_x_ samples, the ultraviolet photoelectron spectroscopy (UPS) technique was employed, as shown in Figure 6. The incident photon energy (hν) was 21.22 eV. As for ZnIn_2_S_4_ (Figure 6a), the secondary electron cutoff energy (E_cutoff_) was 9.32 eV and the Fermi energy (E_Fermi_) was 25.92 eV. The work function of ZnIn_2_S_4_ was calculated to be 3.33 eV using the formula: Work function (WF) = hν + E_cutoff_ − E_Fermi_. Similarly, the work functions for Ti_3_C_2_F_x_, Ti_3_C_2_(OH)_x_ and Ti_3_C_2_O_x_ were calculated to be 4.22 eV, 3.73 eV and 4.57 eV, respectively (Figure 6b–d). Obviously, the work functions of the Ti_3_C_2_T_x_ samples were all higher than that of ZnIn_2_S_4_. Therefore, the photogenerated electrons could transfer from ZnIn_2_S_4_ to Ti_3_C_2_T_x_. Meanwhile, the Schottky barrier could be formed at the ZnIn_2_S_4_/Ti_3_C_2_T_x_ interface due to the difference in the work function and the band alignment between ZnIn_2_S_4_ and Ti_3_C_2_T_x_, which could greatly accelerate the separation and transfer of the photogenerated electrons and holes [34]. The electrostatic potentials of Ti_3_C_2_F_x_, Ti_3_C_2_(OH)_x_ and Ti_3_C_2_O_x_ were obtained from a density functional theory (DFT), as shown in Appendix A. The order of work function values obtained from the DFT calculations was in accordance with that from the UPS characterization. Moreover, the difference in work function between ZnIn_2_S_4_ and Ti_3_C_2_T_x_ was associated with the photogenerated electrons’ transfer ability [35,36]. The largest difference in the work function between ZnIn_2_S_4_ and Ti_3_C_2_O_x_ indicated that Ti_3_C_2_O_x_ showed the strongest electron capture capability from ZnIn_2_S_4_ in the ZnIn_2_S_4_/Ti_3_C_2_O_x_ heterojunction, leading to the significantly high photocatalytic activity.

Based on the aforementioned results, a probable photocatalytic mechanism for ZnIn_2_S_4_/Ti_3_C_2_O_x_ was proposed (Figure 7). The conduction band potential of the parent ZnIn_2_S_4_ was estimated by the Mott-Schottky method (Appendix A). Under visible light irradiation, the photogenerated electrons on the valence band (VB) of ZnIn_2_S_4_ were excited to the conduction band (CB). Because the work function of Ti_3_C_2_O_x_ was higher than that of ZnIn_2_S_4_, photogenerated electrons in the CB of ZnIn_2_S_4_ could quickly migrate to the surface of Ti_3_C_2_O_x_ across the intimate interface, the Schottky junction formed between ZnIn_2_S_4_ and Ti_3_C_2_O_x_ could further prevent the recombination of photogenerated electrons and holes in the ZnIn_2_S_4_/Ti_3_C_2_O_x_. Subsequently, the photogenerated electrons in ZnIn_2_S_4_/Ti_3_C_2_O_x_ were available to react with water to evaluate H_2_, while the holes on the VB of ZnIn_2_S_4_ are consumed by the sacrificial agent TEOA.

## 4. Conclusions

In summary, we have successfully designed and synthesized the surface functionalized Ti_3_C_2_(OH)_x_ and Ti_3_C_2_O_x_ using the surface post-treatments of Ti_3_C_2_F_x_; then Ti_3_C_2_T_x_ (T_x_ = F, OH, O) were employed as the substrate and cocatalysts for the in-situ growth of ZnIn_2_S_4_ to obtain ZnIn_2_S_4_/Ti_3_C_2_T_x_ heterojunctions for photocatalytic H_2_ production. Remarkably, the photocatalytic H_2_ production activity of ZnIn_2_S_4_/Ti_3_C_2_T_x_ was greatly improved, compared to that of ZnIn_2_S_4_. Due to the differences in work function between ZnIn_2_S_4_ and Ti_3_C_2_T_x_, the formation of the Schottky junction could effectively accelerate the separation and migration of photogenerated electrons and holes, and thus boost the photocatalytic H_2_ evolution activity. In particular, among Ti_3_C_2_T_x_ (T_x_ = F, OH, O), the work function of Ti_3_C_2_O_x_ was the largest, and the Ti_3_C_2_O_x_ showed the strongest electron capture ability from ZnIn_2_S_4_. Experimental characterization analyses also demonstrated the rapid separation and transfer of photogenerated electrons and holes of ZnIn_2_S_4_/Ti_3_C_2_O_x_. This work paves the way for photocatalytic applications of MXene-based photocatalysts by tuning their surface termination groups.

## Data Availability

Not applicable.

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
