# Peer review of "Enhanced Photocatalytic Hydrogen Production of ZnIn2S4 by Using Surface-Engineered Ti3C2Tx MXene as a Cocatalyst"

_materials, 2023, doi:10.3390/ma16062168_

Round 1

Reviewer 1 Report

The manuscript discusses the use of a cocatalyst ZnIn2S4/Ti3C2Tx, for photocatalytic H2 production. The authors have prepared three different cocatalysts made ZnIn2S4/Ti3C2Fx, ZnIn2S4/Ti3C2(OH)x, and ZnIn2S4/Ti3C2Ox among them ZnIn2S4/Ti3C2Ox showed better photocatalytic H2 production. The authors have characterized the materials with various characterization techniques and also studied the mechanism of H2 production. The results were interesting. However, the authors have claimed the material is stable but haven’t provided the stability measurements in the manuscript. Hence, the following major concerns should be addressed before its acceptance for publication:

1. The authors should perform the stability measurements in their prepared samples, i.e., authors can perform the photocatalytic H2 production for at least 3 repeated cycles using their prepared samples.

2. Use of Mxenes for photocatalytic H2 production is quite interesting. However, the preparation of Mxenes involves the use of Fluorine-based compounds, which are of huge environmental concern. Does the fluoride ions are being leached during the photocatalysis process? Authors should take the solution of the samples after the photocatalytic process for ICP-MS or AAS analysis to observe the leaching of fluoride ions.

3. Bandgap for the ZnIn2S4/Ti3C2Fx, ZnIn2S4/Ti3C2(OH)x, and ZnIn2S4/Ti3C2Ox should be calculated. 

Author Response

Point 1: The authors should perform the stability measurements in their prepared samples, i.e., authors can perform the photocatalytic H2 production for at least 3 repeated cycles using their prepared samples.

Response 1: Thanks for your advice. We have supplemented the stability test of photocatalytic H2 production over ZnIn2S4/Ti3C2Ox, and the result showed that a slight recession in the photocatalytic activity after 4 cycles, indicating the good photostability of the ZnIn2S4/Ti3C2Ox.

Figure 4c. Recycles test of photocatalytic H2 production over ZnIn2S4/Ti3C2Ox

Point 2: Use of Mxenes for photocatalytic H2 production is quite interesting. However, the preparation of Mxenes involves the use of Fluorine-based compounds, which are of huge environmental concern. Does the fluoride ions are being leached during the photocatalysis process? Authors should take the solution of the samples after the photocatalytic process for ICP-MS or AAS analysis to observe the leaching of fluoride ions.

Response 2: Thanks a lot for your scrupulous comments. ICP-MS or AAS analysis could not be used to detected fluoride ions. The analysis of the solution after the photocatalytic process has been carried out by ion chromatography (IC) to detect the F ions, the result was shown in Figure R1, it can be seen that no F ions was detected, indicating that no F ions was leached during the photocatalysis process. It can be explained as followed: firstly, during the preparation of Ti3C2Fx, Ti3C2Fx product was washed thoroughly with deionized water to remove impurity ions; Secondly, the preparation condition of ZnIn2S4/Ti3C2Fx was stirring at 80 ℃ for 2.5 h, in contrast, the photocatalytic reaction temperature was only 8 ℃, therefore, it was believed that Ti3C2Fx was stable during the photocatalytic process. That is to say, F ions were hardly leached during the photocatalysis process.

Figure R1. a) Chromatogram of NaF (0.5 mg/L) standard solution; b) Chromatogram of solution after photocatlaysis process

Point 3: Bandgap for the ZnIn2S4/Ti3C2Fx, ZnIn2S4/Ti3C2(OH)x, and ZnIn2S4/Ti3C2Ox should be calculated.

Response 3: As suggested by the referee, we have calculated the bandgaps for the ZnIn2S4/Ti3C2Fx, ZnIn2S4/Ti3C2(OH)x, and ZnIn2S4/Ti3C2Ox from the UV-vis DRS spectra using the Tauc’s equation, the results was shown in Figure 3b. The bandgap values estimated from the plots for the samples ZnIn2S4/Ti3C2Fx, ZnIn2S4/Ti3C2(OH)x, and ZnIn2S4/Ti3C2Ox were 2.60 eV, 2.59 eV, and 2.63 eV, respectively.

Figure 3b. Band gaps evaluated from the UV-vis DRS spectra using the Tauc’s equation for the samples ZnIn2S4 and ZnIn2S4/Ti3C2Tx (T= F, OH, O) samples.

Reviewer 2 Report

The manuscript reports the development of efficient co-catalysts, Ti3C2Tx (Tx = F, OH, O) nanosheets for photocatalytic hydrogen production. This photocatalytic activity is further enhanced by insitu growth fabrication of ZnIn2S4 nanosheets on the surface of Ti3C2Ox. The work is really very interesting in terms of the activity of the catalyst and also the tunability that has been achieved through surface modification of the co-catalyst structure. I recommend publication of the article with minor comments to further re-investigate certain issues.

1. It is not exactly clear whether the F terminating groups of Ti3C2Fx have been totally removed during the alkalization treatment with NaOH. If not, then the composition Ti3C2(OH)x is wrongly written and should be rectified based on the residual F composition.

2. Where is the composition of O (23.78%) coming from for Ti3C2Fx, as indicated in the SI? The exact composition should be determined.

3. The binding energy for Ti-OH and Ti-F are different and therefore, if the terminating groups are actually replaced there should be an indication of that through peak shifting in the XPS. The integrity of the crystal structure is still maintained as revealed through the powder patterns of all three materials. The authors should provide a proper explanation supporting the formation of the newer terminating groups.

4. Additionally, XPS reveals only the surface composition. Authors should try to collect ICP-AES to predict the bulk composition of these samples.

5. The authors should provide an explanation that mechanistically support the order of photocatalytic activity in the materials instead of only demonstrating the experimental results.

Author Response

Point 1: It is not exactly clear whether the F terminating groups of Ti3C2Fx have been totally removed during the alkalization treatment with NaOH. If not, then the composition Ti3C2(OH)x is wrongly written and should be rectified based on the residual F composition.

Response 1: Thanks a lot for your scrupulous comments. Actually, the terminating groups F in Ti3C2Fx were extremely difficult to be totally removed during the alkalization treatment. In our manuscript, the Ti3C2Tx (T= F, OH, O) only represented a higher density of Tx on the surface and the exact surface were still a mixture of termination groups F, OH and O bonded with Ti, which was confirmed by the XPS results, it can be seen that the coexistence of Ti-F, Ti-OH and Ti-O bonds in all the three Ti3C2Tx samples.

Point 2: Where is the composition of O (23.78%) coming from for Ti3C2Fx, as indicated in the SI? The exact composition should be determined.

Response 2: We are very grateful for your comments. The composition of O (23.78%) mainly came from the Ti-OH bond, Ti-O bond and the tiny proportion of surface oxidation TiO2. The approximate composition of the termination groups can be calculated from the XPS results, as shown in Table R1.

Table R1. The concentration of the termination groups Tx obtained from XPS.

sample

   F (at %)

          O (at %)

    OH (at %)

   Ti-O

Oxides TiO2

C-OH

Ti-OH

Ti3C2Fx

14.73

   8.20

    2.53

  5.54

  7.51

Ti3C2(OH)x

7.07

   9.01

    3.88

  11.9

  4.02

Ti3C2Ox

7.5

  14.63

    4.01

  8.89

  4.21

Point 3: The binding energy for Ti-OH and Ti-F are different and therefore, if the terminating groups are actually replaced there should be an indication of that through peak shifting in the XPS. The integrity of the crystal structure is still maintained as revealed through the powder patterns of all three materials. The authors should provide a proper explanation supporting the formation of the newer terminating groups.

Response 3: Thanks for your comments. The surface functionalization treatments just modulated the termination groups without changing the crystalline structure of Ti3C2Fx. The structure schematic of Ti3C2Tx nanosheet was shown in Figure R2. It clearly showed that the main structure of Ti3C2Tx was Ti3C2. According to the literatures (ChemSusChem, 2019, 12, 1368-1; J Mater. Chem. A. 2018, 6, 24031-24035; Nano Energy. 2020, 72, 104681), the termination groups Tx in Ti3C2Tx had no influence on their XPS peaks shifting and their crystal structure. In our manuscript, Ti3C2Tx (T= F, OH, O) only represented a higher density of Tx on the surface and the exact surface were still a mixture of termination groups F, OH and O bonded with Ti. After alkalization treatment with NaOH, the termination groups -F was partially replaced by -OH, which was confirmed by XPS and FTIR characterization. And approximate composition of the termination groups can be calculated from the XPS results (Table R1).

Table R1. The concentration of the termination groups Tx obtained from XPS.

sample

   F (at %)

          O (at %)

    OH (at %)

   Ti-O

Oxides TiO2

C-OH

Ti-OH

Ti3C2Fx

14.73

   8.20

    2.53

  5.54

  7.51

Ti3C2(OH)x

7.07

   9.01

    3.88

  11.9

  4.02

Ti3C2Ox

7.5

  14.63

    4.01

  8.89

  4.21

Point 4: Additionally, XPS reveals only the surface composition. Authors should try to collect ICP-AES to predict the bulk composition of these samples.

Response 4: Thanks for your comments. Unfortunately, the F, O and C elements could not be detected by ICP-AES analysis. We tried to analyze the bulk composition of the Ti3C2Ox sample by SEM-EDX spectroscopy. As shown in Figure R2, the investigated EDX exhibited that the as-prepared Ti3C2Ox was composed of Ti, C, O, and F, and Ti and C are main elements in the Ti3C2Ox nanosheets. The O/OH and F elements mainly exist in the terminal groups which was agreement with the previous study [ChemSusChem, 2019, 12, 1368-1; J Mater. Chem. A. 2018, 6, 24031-24035; Nano Energy. 2020, 72, 104681].

Figure R2. EDX analysis of Ti3C2Ox nanosheets

Point 5: The authors should provide an explanation that mechanistically support the order of photocatalytic activity in the materials instead of only demonstrating the experimental results.

Response 5: In our manuscript, the photocatalytic activity part was invertigated firstly, then fluorescence property and photoelectrochemaical characterizations were caaried out to shed light on the fundamental reasons for the order of photocatalytic activity. From the study of structure-activity relationship, it can be seen that the order of photogenerated carriers’ transfer and separation ability is consistent with that of the photocatalytic activity. The ZnIn2S4/Ti3C2Ox with the largest difference in the work function between ZnIn2S4 and Ti3C2Ox, exhibited the fastest transfer and separation ability of photogenerated electrons and holes, further resulting in the excellent photocatalytic H2 production performance.

Round 2

Reviewer 1 Report

The authors have performed sufficient revisions and hence this manuscript could be accepted for publication in the present form.